

# ECOVNet: a highly effective ensemble based deep learning model for detecting COVID-19

Nihad Karim Chowdhury[1], Muhammad Ashad Kabir[2], Md. Muhtadir Rahman[1] and Noortaz Rezoana[1]

[1] Department of Computer Science and Engineering, University of Chittagong, Chittagong, Bangladesh
[2] School of Computing and Mathematics, Charles Sturt University, NSW, Australia

## ABSTRACT

The goal of this research is to develop and implement a highly effective deep learning model for detecting COVID-19. To achieve this goal, in this paper, we propose an ensemble of Convolutional Neural Network (CNN) based on EfficientNet, named ECOVNet, to detect COVID-19 from chest X-rays. To make the proposed model more robust, we have used one of the largest open-access chest X-ray data sets named COVIDx containing three classes—COVID-19, normal, and pneumonia. For feature extraction, we have applied an effective CNN structure, namely EfficientNet, with ImageNet pre-training weights. The generated features are transferred into custom fine-tuned top layers followed by a set of model snapshots. The predictions of the model snapshots (which are created during a single training) are consolidated through two ensemble strategies, i.e., hard ensemble and soft ensemble, to enhance classification performance. In addition, a visualization technique is incorporated to highlight areas that distinguish classes, thereby enhancing the understanding of primal components related to COVID-19. The results of our empirical evaluations show that the proposed ECOVNet model outperforms the state-of-the-art approaches and significantly improves detection performance with 100% recall for COVID-19 and overall accuracy of 96.07%. We believe that ECOVNet can enhance the detection of COVID-19 disease, and thus, underpin a fully automated and efficacious COVID-19 detection system.

## INTRODUCTION

Corona virus disease 2019 (COVID-19) is a contagious disease that was caused by the Severe Acute Respiratory Syndrome Coronavirus 2 (SARS-CoV-2). The disease was first detected in Wuhan City, Hubei Province, China in December 2019, and was related to contact with a seafood wholesale market and quickly spread to all parts of the world (*World Health Organization, 2020*). The World Health Organization (WHO) promulgated the outbreak of the COVID-19 pandemic on 11 March 2020. This perilous virus has not only overwhelmed the world, but also affected millions of lives (*World Health Organization, 2020*). To limit the spread of this infection, all infected countries strive to

Corresponding author
Nihad Karim Chowdhury,
nihad@cu.ac.bd

cover many strategies such as encourage people to maintain social distancing as well as lead hygienic life, enhance the infection screening system through multi-functional testing, seek mass vaccination to reduce the pandemic ahead of time, etc. The reverse transcriptase-polymerase chain reaction (RT-PCR) is a modular diagnosis method, however, it has limitations, such as strict procedures at testing laboratories cause delays to the detection of suspect patients (*Zheng et al., 2020*) and false-negative results may have a greater impact on the prevention and control of the disease (*Fang et al., 2020*).

To make up for the shortcomings of RT-PCR testing, researchers around the world are seeking to promote a fast and reliable diagnostic method to detect COVID-19 infection. The WHO and Wuhan University Zhongnan Hospital issued quick guides (*World Health Organization, 2020*; *Jin et al., 2020a*), suggesting that in addition to detecting clinical symptoms, chest imaging can also be used to evaluate the disease to diagnose and treat COVID-19. In *Rubin et al. (2020)*, the authors have contributed a detailed guideline for medical practitioners to use chest radiography and computed tomography (CT) to screen and assess the disease progression of COVID-19 cases. Although CT scans have higher sensitivity, they also have some drawbacks, such as a high cost and the need for high doses of radiation during screening, which exposes pregnant women and children to greater radiation risks (*Davies, Wathen & Gleeson, 2011*). On the other hand, diagnosis based on chest X-ray appears to be a propitious solution for COVID-19 detection and treatment. *Ng et al. (2020)* reported that the pulmonary manifestation of COVID-19 infection was immensely delineated by chest X-ray images.

The purpose of this study is to ameliorate the accuracy of the COVID-19 detection system from chest X-ray images. In this context, we contemplate a CNN-based architecture, since it is renowned for its excellent recognition performance in image classification and detection. For medical image analysis, higher detection accuracy along with crucial findings is a top aspiration, and in current years, CNN based architectures are comprehensively featured the critical findings related to medical imaging. To achieve the defined purpose, this paper presents a novel CNN-based architecture called ECOVNet, exploiting the cutting-edge EfficientNet (*Tan & Le, 2019*) family of CNN models together with ensemble strategies. The pipeline of the proposed architecture starts with the data augmentation approach, then optimizes and fine-tunes the pre-trained EfficientNet models, creating respective model's snapshots. After that, generated model snapshots are integrated into an ensemble, i.e., soft ensemble and hard ensemble, to make predictions. The motivation for using EfficientNet models is that they are known for their high accuracy, while being smaller and faster than the best existing CNN architectures. Moreover, an ensemble technique has been effective in predicting since it produces a lower error rate compared with the prediction of a single model. The use of ensemble techniques on different CNN models has proven to be an effective technique for image-based diagnosis and biomedical research (*Kumar et al., 2017*). Owing to the limited number of COVID-19 images currently available, diagnosing COVID-19 infection is more challenging, therefore refining a visual explainable approach is applied for further analysis. In this regard, we use a gradient-based class activation mapping algorithm, i.e., Grad-CAM (*Selvaraju et al., 2017*), providing explanations of the predictions and identifying relevant

features associated with COVID-19 infection. The key contributions of this paper are as follows:

- We propose a novel CNN-based architecture that includes pre-trained EfficientNet model for feature extraction and model snapshots to detect COVID-19 from chest X-rays.
- Assuming the decisions of multiple radiologists are considered in the final prediction, we propose an ensemble in the proposed architecture to make predictions, thus making a credible and fair evaluation of the system.
- We visualize a class activation map through Grad-CAM to explain the prediction and identify the critical regions in the chest X-ray.
- We present an empirical evaluation of our model compared with state-of-the-art models to appraise the effectiveness of the proposed architecture in detecting COVID-19.

The remainder of the paper is arranged as follows. "Related Works" discusses related work. "Methodology" explains the details of the dataset and presents the ECOVNet architecture. The results of our empirical evaluation are presented in "Experiments and Results". Finally, "Conclusion and Future Work" concludes the paper and highlights future work.

## RELATED WORKS

Significant improvements achieved through CNN technology have prompted computational medical imaging researchers to study the potential of CNN in medical images obtained through CT, magnetic resonance imaging (MRI), and X-rays. Many researchers have explored CNN technology as an effective way to classify (*Kaur & Gandhi, 2020*) and segment (*Zhang et al., 2015*) MR brain images, and have obtained the best performance. What's more, pneumonia detection using chest X-rays with CNN is a state-of-the-art technology which highlights the potential of image diagnosis systems (*Rajpurkar et al., 2017*). The need to identify COVID-19 infections faster, the latest application areas of CNN-based AI systems are booming, which will potentially speed up the analysis of various medical images. An in-depth survey of the application of CNN technology in COVID-19 detection and automatic lung segmentation using X-rays and CT images is presented in *Shoeibi et al. (2020)*. Due to the need to interpret chest CT images faster, *Jin et al. (2020b)* proposed an AI system based on deep learning that can speed up the analysis of chest CTs to detect COVID-19 and validated using a large multi-class dataset. A new CNN architecture named COVID-Net and a large chest X-ray benchmark dataset (COVIDx) have introduced in *Wang, Lin & Wong (2020)*. The proposed COVID-Net obtained the best test accuracy of 93.3%, and uses an interpretability method to make predictions. *Luz et al. (2020)* proposed a new deep learning framework that extends the EfficientNet (*Tan & Le, 2019*) series, which is well known for its excellent prediction performance and fewer computational steps. Their experimental evaluation showed noteworthy classification performance, especially in COVID-19 cases. A CNN model called DarkCovidNet (*Ozturk et al., 2020*) was proposed for the automatic detection of COVID-19 using chest X-ray images where the proposed method carried out two types

of classification, one for binary classification (such as COVID-19 and No-Findings) and the other for multi-class classification (such as COVID-19, No-Findings and pneumonia). Finally, the authors provided an intuitive explanation through a heat map, so it can assist radiologists to find the affected area on the chest X-ray. Another study (*Karim et al., 2020*) proposed an explainable CNN-based method named DeepCOVIDExplainer to automatically detect COVID-19 cases from chest X-ray images. The method involved adjusting a neural ensemble technique then highlighting class-discriminating regions. *Khan, Shah & Bhat (2020)* proposed a model named CoroNet that used Xception architecture pre-trained on an ImageNet dataset and trained on their benchmark created from two publicly available datasets, and carried out two different classification performance measurements, i.e., three and four classes with classification accuracy of 95% and 89.6%, respectively. In another work, *Mahmud, Rahman & Fattah (2020)* proposed a CNN-based model called CovXNet, which uses depth-wise dilated convolution. At first, the model was trained with some non-COVID-19 pneumonia images; the acquired learning was used with some additional fine-tuning layers that trained again with a smaller number of chest X-rays related to COVID-19 and other pneumonia cases. As features are extracted from different resolutions of X-rays, a stacking algorithm is used in the prediction process for multi-classclassification, the accuracy of CovXNet is 90.3%.

An advanced custom CNN architecture, COVID-Net (*Wang, Lin & Wong, 2020*) was implemented and tested using a large COVID-19 benchmark, but due to the large number of parameters, the computational overhead of this model is high. Another CNN-based modular architecture, named PDCOVIDNet, is proposed by *Chowdhury, Rahman & Kabir (2020)*, which consists of a parallel stack of multi-layer filter blocks in a cascade with a classification and visualization block. The authors reported the model was effective when compared with a number of well-known CNN architectures and showed precision and recall of 96.58% and 96.59%, respectively. Table 1 shows an overview of some CNN-based architectures for detecting COVID-19 from chest X-rays.

Most of the existing studies, as discussed above, make predictions based on the output of a single model, only a few methods (such as *Karim et al. (2020)* and *Mahmud, Rahman & Fattah (2020)*) used an ensemble. The key benefit of the ensemble is that it can reduce prediction errors, thus makes the model more versatile. In *Karim et al. (2020)*, authors used an ensemble on heterogeneous models, i.e., VGG19 (*Simonyan & Zisserman, 2015*), ResNet18 (*He et al., 2016*), and DenseNet161 (*Huang et al., 2017b*), but it has two flaws. Firstly, each model requires a separate training session, and secondly, an individual model suffers from training many parameters. In another method (*Mahmud, Rahman & Fattah, 2020*) used an ensemble on a single model with various image resolutions, and for each image resolution, it creates a separate model and stacks it for prediction, which incurs a significant computational overhead. To address the aforementioned problems, we use a lightweight but effective model EfficientNet- it is 8.4 times smaller and 6.1 times faster than the best existing CNN (*Tan & Le, 2019*). Also, we force large changes in model weights through the recursive learning rate, create model snapshots in the same training, and further apply the ensemble to make the proposed architecture more robust, thereby achieving a higher detection rate compared to other methods.

**Table 1** Overview of CNN based architectures for detecting COVID-19 from chest X-rays.

| Method | Data Source | Architecture | Pre-trained Weight | Ensemble | Visualization |
|---|---|---|---|---|---|
| *Wang, Lin & Wong (2020)* | COVIDx (*Wang, Lin & Wong, 2020*) | COVID-Net | ImageNet | No | GSInquire (*Lin et al., 2019*) |
| *Luz et al. (2020)* | COVIDx | EfficientNet | ImageNet | No | Heat Map |
| *Ozturk et al. (2020)* | *Cohen, Morrison & Dao (2020)*, NIH Chest X-ray dataset (*Wang et al., 2017*) | DarkCovidNet | No | No | Grad-CAM |
| *Karim et al. (2020)* | COVIDx | VGG, ResNet, DenseNet | ImageNet | Yes | Grad-CAM, Grad-CAM++ (*Chattopadhay et al., 2018*), LRP (*Bach et al., 2015*) |
| *Khan, Shah & Bhat (2020)* | *Cohen, Morrison & Dao (2020)*, P. Mooney (*Mooney, 2017*) | Xception | ImageNet | No | No |
| *Mahmud, Rahman & Fattah (2020)* | Mendeley Data, V2 (*Kermany, Zhang & Goldbaum, 2018*), 305 COVID-19 images | CovXNet | Non-COVID X-rays | Yes | Grad-CAM |
| *Chowdhury, Rahman & Kabir (2020)* | COVID-19 Radiography Database (*Chowdhury et al., 2020*) | PDCOVIDNet | No | No | Grad-CAM, Grad-CAM++ |

# METHODOLOGY

In this section, we briefly discuss our approach. First, we describe the benchmark dataset and data augmentation strategy used in the proposed architecture. Next, we outline the proposed ECOVNet architecture, including network construction using a pre-trained EfficientNet model and training methods, and then model ensemble strategies. Finally, to make disease detection more acceptable, we integrate decision visualizations to highlight pivotal facts with visual markers.

## Dataset

In this sub-section, we concisely inaugurate the benchmark dataset, named COVIDx (*Wang, Lin & Wong, 2020*), that we used in our experiment. The dataset comprises three categories of images—COVID-19, normal and pneumonia, with a total of 13,914 images for training and 1,579 for testing (accessed on 17 July 2020). To generate the COVIDx, the authors (*Wang, Lin & Wong, 2020*) used five different publicly accessible data repositories:

- COVID-19 Image Data Collection (*Cohen, Morrison & Dao, 2020*)—non-COVID-19 pneumonia and COVID-19 cases are taken from this repository.
- COVID-19 Chest X-ray Dataset Initiative (*Chung, 2020b*)—only COVID-19 cases are taken from this repository.
- ActualMed COVID-19 Chest X-ray Dataset Initiative (*Chung, 2020a*)—only COVID-19 cases are taken from this repository.
- Radiological Society of North America (RSNA) Pneumonia Detection Challenge dataset (*RSNA, 2019*)—normal and non-COVID-19 pneumonia cases.

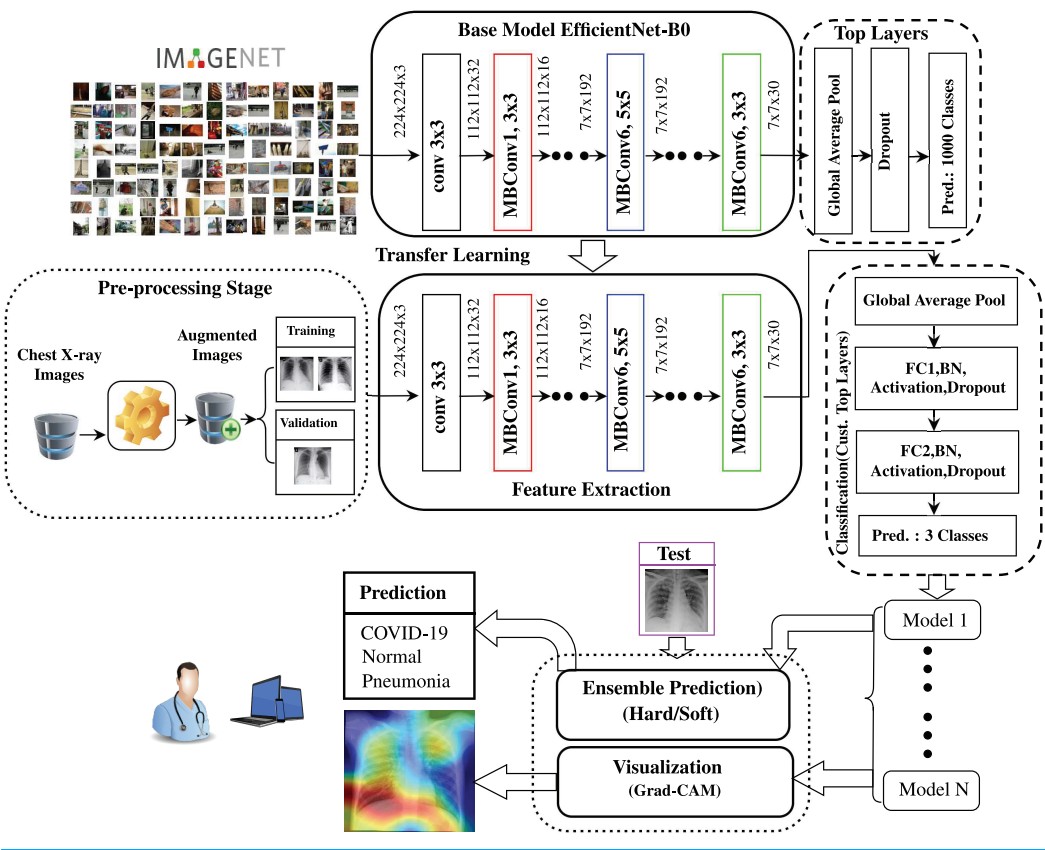

**Figure 1** Graphical representation of the proposed ECOVNet architecture.

- COVID-19 Radiography Database (*Chowdhury et al., 2020*)—only COVID-19 cases.

## Data augmentation

Data augmentation is usually performed during the training process to expand the training set. As long as the semantic information of an image is preserved, the transformation can be used for data augmentation. Using data augmentation, the performance of the model can be improved by solving the problem of overfitting. Although the CNN model has properties such as partial translation-invariant, augmentation strategies (i.e., translated images) can often considerably enhance generalization capabilities (*Goodfellow, Bengio & Courville, 2016*). Data augmentation strategies provide various alternatives, each of which has the advantage of interpreting images in multiple ways to present important features, thereby improving the performance of the model. We have considered the following transformations for augmentation: horizontal flip, rotation, shear, and zoom.

## ECOVNet architecture

Figure 1 shows a graphical presentation of the proposed ECOVNet architecture using a pre-trained EfficientNet model. After augmenting the COVIDx dataset, we used the

pre-trained EfficientNet model (*Tan & Le, 2019*) as a feature extractor. This step ensures that the pre-trained EfficientNet model can extract and learn useful chest X-ray features, and can generalize it well. Indeed, EfficientNets are an order of models that are obtained from a base model, i.e., EfficientNet-B0. In Fig. 1, we depicted our proposed architecture using EfficientNet-B0 for the sake of brevity, however, during the experimental evaluation, we have also considered five EfficientNets B1 to B5. The output features from the pre-trained model fed into our proposed custom top layers through two fully connected layers, which are integrated with batch normalization, activation, and dropout. We generated several snapshots in a training session, and then combined their predictions with an ensemble prediction. At the same time, the visualization approach, which can qualitatively analyze the relationship between input examples and model predictions, was incorporated into the following part of the proposed model.

### Pre-trained efficientnet feature extraction

EfficientNets are a series of models (namely EfficientNet-B0 to B7) that are derived from the baseline network (often called EfficientNet-B0) by scaling it up. By adopting a compound scaling method in all dimensions of the network, i.e., width, depth, and resolution, EfficientNets have attracted attention due to their superior prediction performance. The intuition of using compound scaling is that scaling any dimension of the network (such as width, depth, or image resolution) can increase accuracy, but for larger models, the accuracy gain will decrease. To scale the dimensions of the network systematically, compound scaling uses a compound coefficient that controls how many more resources are functional for model scaling, and the dimensions are scaled by the compound coefficient in the following way (*Tan & Le, 2019*):

$$
\begin{aligned}
depth : d &= \alpha^{\phi} \\
width : w &= \beta^{\phi} \\
resolution : r &= \gamma^{\phi} \\
s.t.\ \alpha . \beta^2 . \gamma^2 &\approx 2 \\
\alpha \geq 1, \beta &\geq 1, \gamma \geq 1
\end{aligned} \tag{1}
$$

where $\phi$ is the compound coefficient, and $\alpha$, $\beta$, and $\gamma$ are the scaling coefficients of each dimension that can be fixed by a grid search. After determining the scaling coefficients, they are applied to the baseline network (EfficientNet-B0) for scaling to obtain the desired target model size. For instance, in the case of EfficientNet-B0, when $\phi = 1$ is set, the optimal values are yielded using a grid search, i.e., $\alpha = 1.2$, $\beta = 1.1$, and $\gamma = 1.15$, under the constraint of $\alpha . \beta^2 . \gamma^2 \approx 2$ (*Tan & Le, 2019*).

The feature extraction of the EfficientNet-B0 baseline architecture comprises several mobile inverted bottleneck convolution (MBConv) (*Sandler et al., 2018*; *Tan et al., 2019*) blocks with built-in squeeze-and-excitation (SE) (*Hu, Shen & Sun, 2018*), batch normalization, and swish activation (*Ramachandran, Zoph & Le, 2017*) as integrated into EfficientNet. Table 2 shows the detailed information of each layer of the EfficientNet-B0 baseline network. EfficientNet-B0 consists of 16 MBConv blocks varying in several aspects, for instance, kernel size, feature maps expansion phase, reduction ratio, etc.

**Table 2 EfficientNet-B0 baseline network layers outline.**

| Stage | Operator | Resolution | #Output Feature Maps | #Layers |
|---|---|---|---|---|
| 1 | Conv 3 × 3 | 224 × 224 | 32 | 1 |
| 2 | MBConv1, k3 × 3 | 112 × 112 | 16 | 1 |
| 3 | MBConv6, k3 × 3 | 112 × 112 | 24 | 2 |
| 4 | MBConv6, k5 × 5 | 56 × 56 | 40 | 2 |
| 5 | MBConv6, k3 × 3 | 28 × 28 | 80 | 3 |
| 6 | MBConv6, k5 × 5 | 14 × 14 | 112 | 3 |
| 7 | MBConv6, k5 × 5 | 14 × 15 | 192 | 4 |
| 8 | MBConv6, k3 × 3 | 7 × 7 | 320 | 1 |
| 9 | Conv 1 × 1 & Pooling & FC | 7 × 7 | 1280 | 1 |

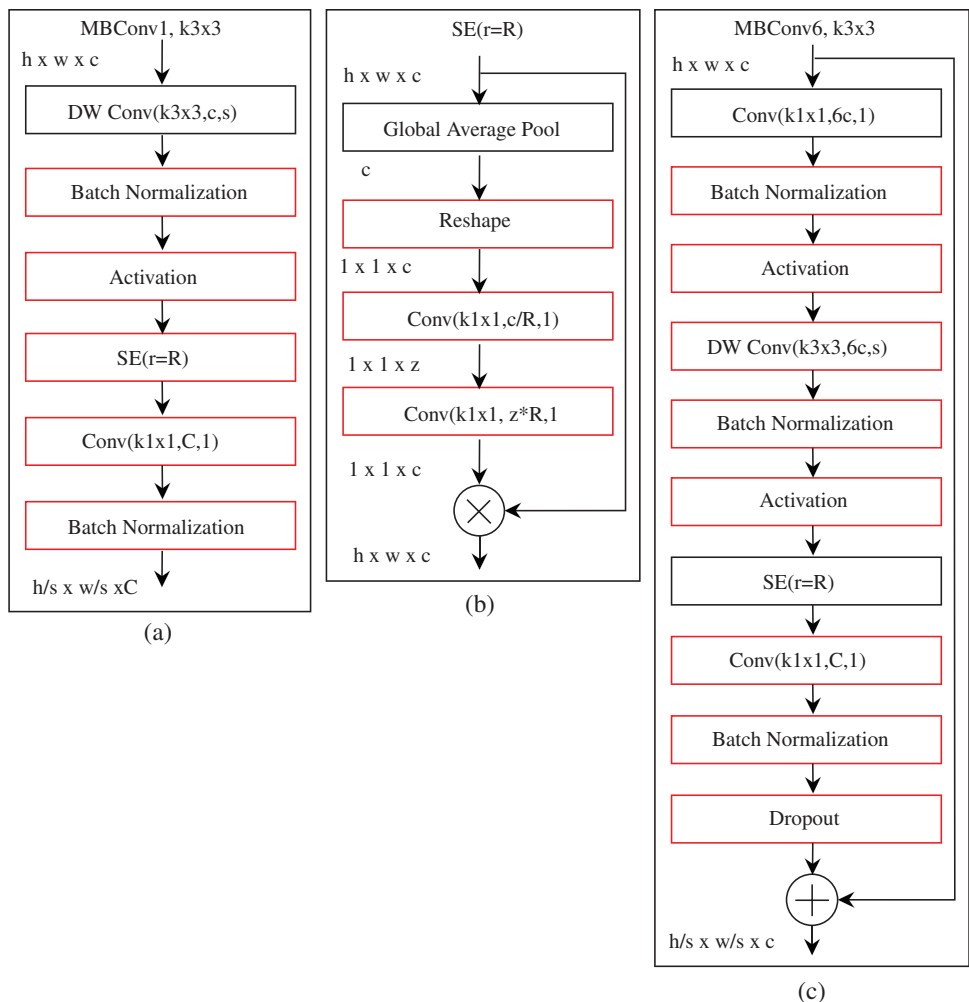

**Figure 2 The basic building block of EfficientNet-B0.** All MBConv blocks take the height, width, and channel of h, w, and c as input. C is the output channel of the two MBConv blocks. (Note that, MBConv, mobile inverted bottleneck convolution; DW Conv, depth-wise convolution; SE, squeeze-excitation; Conv, convolution). Here, (A) and (C) are mobile inverted bottleneck convolution blocks, but the difference is that (C) is six times that of (A). (B) is squeeze-excitation block.

A complete workflow of the MBConv1, k3 × 3 and MBConv6, k3 × 3 blocks is shown in Fig. 2. Both MBConv1, k3 × 3 and MBConv6, k3 × 3 use depthwise convolution, which integrates a kernel size of 3 × 3 with the stride size of *s*. In these two blocks, batch normalization, activation, and convolution with a kernel size of 1 × 1 are integrated. The skip connection and a dropout layer are also incorporated in MBConv6, k3 × 3, but this is not the case with MBConv1, k3 × 3. Furthermore, in the case of the extended feature map, MBConv6, k3 × 3 is six times that of MBConv1, k3 × 3, and the same is true for the reduction rate in the SE block, that is, for MBConv1, k3 × 3 and MBConv6, k3 × 3, *r* is fixed to 4 and 24, respectively. Note that, MBConv6, k5 × 5 performs the identical operations as MBConv6, k3 × 3, but MBConv6, k5 × 5 applies a kernel size of 5 × 5, while a kernel size of 3 × 3 is used by MBConv6, k3 × 3.

Instead of random initialization of network weights, we instantiate ImageNet's pre-trained weights in the EfficientNet model thereby accelerating the training process. ImageNet has performed a great feat in the field of image analysis, since it is composed of more than 14 million images covering eclectic classes. The rationale for using pre-trained weights is that the imported model already has sufficient knowledge in the broader aspects of the image domain. As shown in several studies (*Rajaraman et al., 2020*; *Narin, Ceren & Ziynet, 2020*), there is reason for optimism in using pre-trained ImageNet weights in state-of-the-art CNN models even when the problem area (namely COVID-19 detection) is considerably distinct from the one in which the original weights were obtained. The optimization process will fine-tune the initial pre-training weights in the new training phase so that we can fit the pre-trained model to a specific problem domain, such as COVID-19 detection. For the feature extraction process of the proposed ECOVNet architecture applying pre-trained ImageNet weights is executed after the image augmentation process, as presented in Fig. 1.

### Classifier

After the feature extraction process, a customized top layer is used, which works as the classifier shown in Fig. 1. The output features of the pre-trained EfficientNet model turns out as globally averaged. To perform the classification task, we used a two-layer MLP (usually called a fully connected (FC) layer), which captures the globally averaged features of EfficientNet model through two neural layers (each neural layer has 512 nodes). In between FC layers, we included a batch normalization, activation, and dropout layer. Batch normalization greatly accelerates the training of deep networks and increases the stability of neural networks (*Ioffe & Szegedy, 2015*). It makes the optimization process smoother, resulting in a more predictable and stable gradient behavior, thereby speeding up training (*Santurkar et al., 2018*). In this study, for activation function, we have preferred Swish (*Ramachandran, Zoph & Le, 2017*), which is defined as:

$$f(x) \quad = x \cdot \sigma(x) \tag{2}$$

where $\sigma(x) = (1 + exp(-x))^{-1}$ is the sigmoid function. Swish consistently outperforms other activation functions including Rectified Linear Unit (ReLU) (*Nair & Hinton, 2010*), which is the most successful and widely-used activation function, on deep networks

applied to a variety of challenging fields including image classification and machine translation. Swish has many characteristics, such as one-sided boundedness at zero, smoothness, and non-monotonicity, which play an important role in improving it (*Ramachandran, Zoph & Le, 2017*). After performing the activation operation, we integrated a dropout (*Srivastava et al., 2014*) layer, which is one of the preeminent regularization methods to reduce overfitting and make better predictions. This layer can randomly drop certain FC layer nodes, which means removing all randomly selected nodes, along with all its incoming and outgoing weights. The number of randomly selected nodes drop in each layer is obtained with a probability $p$ independent of other layers, where $p$ can be chosen by using either a validation set or a random estimate (i.e., $p = 0.5$). In this study, we maintained a dropout size of 0.3. Next, the classification layer used the softmax activation function to render the activation from the previous FC layers into a class score to determine the class of the input chest X-ray image as COVID-19, normal, and pneumonia. The softmax activation function is defined as follows:

$$s(y_i) = \frac{e^{y_i}}{\sum_{j=1}^{C} e^{y_j}} \tag{3}$$

where $C$ is the total number of classes. This normalization limits the output sum to 1, so the softmax output $s(y_i)$ can be interpreted as the probability that the input belongs to the $i$ class. In the training process, we apply the categorical cross-entropy loss function, which uses the softmax activation function in the classification layer to measure the loss between the true probability of the category and the probability of the predicted category. The categorical cross-entropy loss function is defined as

$$l = -\sum_{n=1}^{N} \log \left( \frac{e^{y_{i,n}}}{\sum_{j=1}^{C} e^{y_{j,n}}} \right). \tag{4}$$

The total number of input samples is denoted as $N$, and $C$ is the total number of classes, that is, $C = 3$ in our case.

### Model snapshots and ensemble prediction

The main concept of building model snapshots is to train one model with constantly reducing the learning rate to attain a local minimum and save a snapshot of the current model's weight. Later, it is necessary to actively increase the learning rate to retreat from the current local minimum requirements. This process continues repeatedly until it completes cycles. One of the main methods for creating model snapshots for CNN is to collect multiple models during a single training run with cyclic cosine annealing (*Huang et al., 2017a*). The cyclic cosine annealing method starts from the initial learning rate, then gradually decreases to the minimum, and then rapidly increases. The learning rate of cyclic cosine annealing in each epoch is defined as:

$$\alpha(t) \quad = \frac{\alpha_0}{2}\left(\cos\left(\frac{\pi \mathrm{mod}(t-1, \lceil T/M \rceil)}{\lceil T/M \rceil}\right) + 1\right) \tag{5}$$

where $\alpha(t)$ is the learning rate at epoch $t$, $\alpha_0$ is the initial learning rate, $T$ is the total number of training iterations and $M$ is the number of cycles. The weight at the bottom of each cycle is regarded as the weight of the snapshot model. The following learning rate cycle uses these weights, but allows the learning algorithm to converge to different solutions, thereby generating diverse snapshots model. After completing $M$ cycles of training, we get $M$ model snapshots $s_1 \ldots s_M$, each of which will be utilized in the ensemble prediction.

An ensemble through model snapshots is more effective than a structure based on a single model only. Therefore, compared with the prediction of a single model, the ensemble prediction reduces the generalization error, thereby improving the prediction performance. We have experimented with two ensemble strategies, i.e., *hard ensemble* and *soft ensemble*, to consolidate the predictions of snapshots model to classify chest X-ray images as COVID-19 or normal or pneumonia. Both hard ensemble and soft ensemble use the last $m$ ($m \leq M$) model's softmax outputs since these models have a tendency to have the lowest test error. We also consider class weights to obtain a softmax score before applying the ensemble. Let $O_i(x)$ be the softmax score of the test sample $x$ of the $i$-th snapshot model. Using hard ensemble, the prediction of the $i$-th snapshot model is defined as

$$H_i \quad = argmax_x O_i(x). \tag{6}$$

The final ensemble constrains to aggregate the votes of the classification labels (i.e., COVID-19, normal, and pneumonia) in the other snapshot models and predict the category with the most votes. On the other hand, the output of the soft ensemble includes averaging the predicted probabilities of class labels in the last $m$ snapshots model defined as

$$S \quad = \frac{1}{m}\sum_{i=0}^{m-1} O_{M-i}(x). \tag{7}$$

Finally, the class label with the highest probability is used for the prediction. The creation of model snapshots and ensemble predictions are integrated at the end of the proposed architecture, as shown in Fig. 1.

### Hyper-parameters adjustment

Fine-tuned hyper-parameters have a great impact on the performance of the model because they directly govern the training of the model. What's more, fine-tuned parameters can avoid overfitting and form a generalized model. Since we have dealt with an unbalanced dataset, the proposed architecture may have potential to confront the problem of overfitting. In order to solve the problem of overfitting, we use L1L2 weight decay regularization with coefficients 1e−5 and 1e−3 in FC layers. Next, dropout is another successful regularization technique that has been integrated into the proposed

architecture, especially in FC layers with $p = 0.3$, to suppress overfitting. In the experiments on the proposed architecture, we have explored the Adam optimizer (*Kingma & Ba, 2014*), which can converge faster. When creating snapshots, we set the number of epochs to 25, the minimum batch size to 8, the initial learning rate to 1e−4, and the number of cycles to 5, thus providing 5 snapshots for each model, on which we build up the ensemble prediction.

### *Visual explanations using grad-CAM*

Although the CNN-based modular architecture provides encouraging recognition performance for image classification, there are still several issues where it is challenging to reveal why and how to produce such impressive results. Due to its black-box nature, it is sometimes contrary to apply it in a medical diagnosis system where we need an interpretable system, i.e., visualization as well as an accurate diagnosis. Although it has certain challenges, researchers are still trying to develop an efficient visualization technique. Such a technique would benefit the health-care system by assisting medical practitioners to distinguish correlations and patterns in imaging, and allowing them to perform more efficacious data analysis. In the field of detecting COVID-19 through chest X-rays, some early studies focused on visualizing the behavior of CNN models to distinguish between different categories (such as COVID-19, normal, and pneumonia), so they can produce explanatory models. In our proposed model, we applied a gradient-based approach named Grad-CAM (*Selvaraju et al., 2017*), which measures the gradients of features maps in the final convolution layer on a CNN model for a target image, to foreground the critical regions that are class-discriminating saliency maps. In Grad-CAM, gradients that are flowing back to the final convolutional layer in a CNN model are globally averaged to calculate the target class weights of each filter. A Grad-CAM heat-map is a combination of weighted feature maps, followed by a ReLU activation. The class-discriminative saliency map $L^c$ for the target image class $c$ is defined as follows(*Selvaraju et al., 2017*):

$$L_{i,j}^c = \text{ReLU}(\sum_k w_k^c A_{i,j}^k), \tag{8}$$

where $A_{i,j}^k$ denotes the activation map for the $k$-th filter at a spatial location $(i,j)$, and ReLU captures the positive features of the target class. The target class weights of the $k$-th filter is computed as:

$$w_k^c = \frac{1}{Z} \sum_i \sum_j \frac{\partial Y^c}{\partial A_{i,j}^k}, \tag{9}$$

where $Y^c$ is the probability of classifying the target category as $c$, and the total number of pixels in the activation map is denoted as $Z$. The Grad-CAM visualization of each model snapshot is incorporated at the end edge of the proposed architecture, as displayed in Fig. 1.

## EXPERIMENTS AND RESULTS

In this section, we evaluate the classification performance of our proposed ECOVNet model and compare it's performance with state-of-the-art methods. We consider several

**Table 3  Image partition of training, validation and testing set.**

| Category | COVID-19 | Normal | Pneumonia | Total |
|---|---|---|---|---|
| Training | 441 | 7,170 | 4,914 | 12,525 |
| Validation | 48 | 796 | 545 | 1,389 |
| Testing | 100 | 885 | 594 | 1,579 |

**Table 4  Image resolution and total number of parameters of ECOVNet considering the base models of EfficientNet (B0 to B5).**

| Base Model | Image Resolution | Parameter Size (ECOVNet) |
|---|---|---|
| EfficientNet-B0 | 224 × 224 | 4,978,847 |
| EfficientNet-B1 | 240 × 240 | 7,504,515 |
| EfficientNet-B2 | 260 × 260 | 8,763,893 |
| EfficientNet-B3 | 360 × 360 | 11,844,907 |
| EfficientNet-B4 | 380 × 380 | 18,867,291 |
| EfficientNet-B5 | 456 × 456 | 29,839,091 |

experimental settings to analyze the robustness of the ECOVNet model. All our programs are written in Python, and the software pile is composed of Keras with TensorFlow and scikit-learn. The source code and models are publicly available in GitHub (https://github.com/nihad8610/ECOVNet).

## Dataset and parameter settings

In this section, we introduce the distribution of the benchmark dataset and the model parameters generated in the experiment. We used COVIDx (*Wang, Lin & Wong, 2020*) which is one of the largest publicly available COVID-19 dataset. This dataset comprises three categories of images—COVID-19, normal and pneumonia and has come with a separate training set (13,914 images) and testing set (1,579 images). We further split the training set into training and validation with a ratio of 9:1. We used the original test set unchanged. The entire image distribution of training, validation, and testing is shown in Table 3.

In our experiment, we use EfficientNet B0 to B5 as base models. However, the input image resolution size is different for each base model, and the size increases from B0 to B5. As the image resolution size increases, the model needs more layers to capture the finer-grained patterns, thereby increasing the size of the parameter in the model. Table 4 displays a list of input image resolution size for each base model and the total number of parameters generated during training.

## Evaluation metrics

In order to evaluate the performance of the proposed method, we considered the following evaluation metrics: accuracy, precision, recall, F1-score, confidence interval (CI), receiver

operating characteristic (ROC) curve and area under the curve (AUC). The definitions of accuracy, precision, recall and F1 score are as follows:

$$\text{Accuracy} = \frac{TP + TN}{\text{Total Samples}} \tag{10}$$

$$\text{Precision} = \frac{TP}{TP + FP} \tag{11}$$

$$\text{Recall} = \frac{TP}{TP + FN} \tag{12}$$

$$\text{F1} = 2 \times \frac{\text{Precision} \times \text{Recall}}{\text{Precision} + \text{Recall}} \tag{13}$$

where TP stands for true positive, and TN, FP, and FN stand for true negative, false positive, and false negative, respectively. Since the benchmark dataset is not balanced, the F1 score may be a more substantial evaluation metric. For example, COVID-19 has 589 images and non-COVID-19, that is, normal and pneumonia have 8,851 and 6,053 images, respectively. Moreover, a 95% CI is considered as it is a more practical metric compared with specific performance indicators. It can increase the level of statistical significance and can reflect the reliability of the problem domain. Finally, we displayed the ROC curve to display the results and measured the area under the ROC curve (usually called AUC) to provide information about the effectiveness of the model. The ROC curve is plotted between True Positive Rate (TPR)/Recall and False Positive Rate (FPR), with FPR defined as:

$$\text{FPR} = \frac{FP}{FP + TN}. \tag{14}$$

## Prediction performance

Table 5 reports the prediction performances of the proposed ECOVNet model without using ensemble. The results show that ECOVNet with EfficientNet-B5 pre-trained weights outperforms other base models for the case of images with and without augmentation. It shows that feature extraction using an optimized model that considers three aspects, namely higher depth and width, and a broader image resolution, can capture more and finer details, thereby improving classification accuracy. Without augmentation, under the condition of without ensemble, ECOVNet's accuracy reaches 96.26%, and its performance is lower for with augmentation, reaching 94.68% accuracy. We calculated accuracy with 95% CI. A tight range of CI means higher precision, while the wide range of CI indicates the opposite. As we can see, the CI interval is in a narrow range for the case of no augmentation, and the CI range is wider for the case of augmentation. Furthermore, Fig. 3 shows the training loss of ECOVNet considering EfficientNet-B5.

We implement two ensemble strategies: hard ensemble and soft ensemble. Each ensemble has a total of 5 model snapshots that are generated during a single training. Tables 6 and 7 show the classification results using ensembles without augmentation and with augmentation, respectively. As shown in Table 6, in handling COVID-19 cases,
**Table 5 Prediction performance of proposed ECOVNet without using ensemble.**

| Method | Pre-trained Weight | Precision(%) | Recall(%) | F1-score(%) | Accuracy(%)(95% CI) |
|---|---|---|---|---|---|
| ECOVNet (Without Augmentation) | EfficientNet-B0 | 93.27 | 93.29 | 93.27 | 93.29± 1.23 |
| | EfficientNet-B1 | 94.28 | 94.30 | 94.26 | 94.30± 1.14 |
| | EfficientNet-B2 | 93.24 | 93.03 | 93.08 | 93.03± 1.26 |
| | EfficientNet-B3 | 95.56 | 95.57 | 95.56 | 95.57± 1.01 |
| | EfficientNet-B4 | 95.52 | 95.50 | 95.50 | 95.50± 1.02 |
| | EfficientNet-B5 | **96.28** | **96.26** | **96.26** | **96.26 ± 0.94** |
| ECOVNet (With Augmentation) | EfficientNet-B0 | 91.71 | 74.10 | 79.72 | 74.10± 2.16 |
| | EfficientNet-B1 | 91.02 | 86.19 | 87.67 | 86.19± 1.70 |
| | EfficientNet-B2 | 93.60 | 93.10 | 93.24 | 93.10± 1.25 |
| | EfficientNet-B3 | 92.60 | 90.25 | 90.92 | 90.25± 1.46 |
| | EfficientNet-B4 | 94.32 | 93.73 | 93.89 | 93.73± 1.20 |
| | EfficientNet-B5 | **94.79** | **94.68** | **94.70** | **94.68 ± 1.11** |

Note:
**Bold** indicates that the method has statistically better performance than other methods.

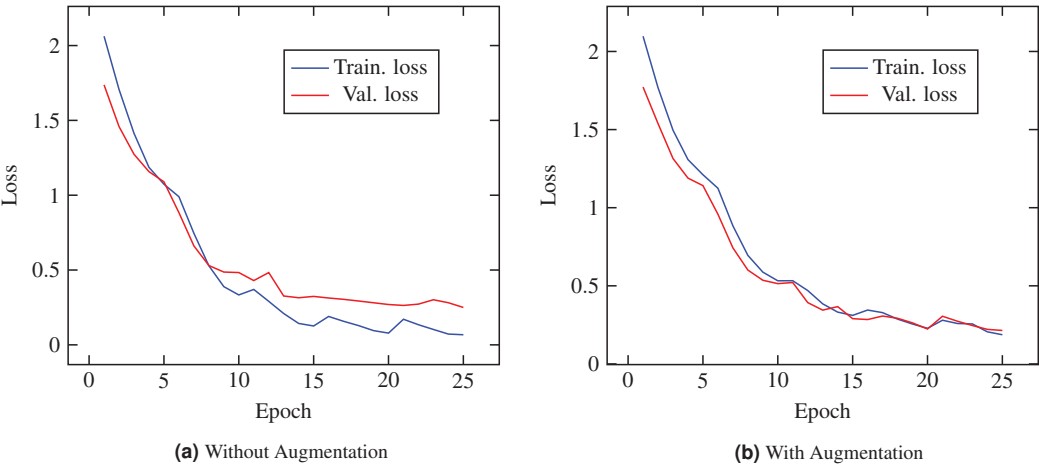

**(a)** Without Augmentation          **(b)** With Augmentation

**Figure 3 Loss curve of ECOVNet (Base model EfficientNet-B5) during training.** (A) Shows characteristics loss curve while augmentation is not applied during the training process. (B) Represents characteristics of the loss curve when augmentation is applied.

the ensemble methods are significantly better than the no ensemble method. More specifically, the recall hits its maximum value of 100%, and to a large extent, this result demonstrates the robustness of our proposed architecture. Furthermore, for COVID-19 detection, soft ensemble seems to be the preferred method due to its recall and F1-score of 100% and 96.15%, respectively. In the soft ensemble, the average softmax score of each category affects the direction of the desired result, thus the performance of the soft ensemble is better than the hard ensemble. Owing to the uneven distribution of the test set, an F1-score may be more reliable than an accuracy.

For augmentation, we see that the ensemble methods present better results than the no ensemble (see Table 7). When comparing the two ensemble methods, we see that hard

**Table 6  Class-wise classification results of ECOVNet (Base model EfficientNet-B5) without augmentation.**

| Method | Class | Precision (%) | Recall (%) | F1-score (%) | Accuracy (%)(95% CI) |
|---|---|---|---|---|---|
| ECOVNet (Without Ensemble) | COVID-19 | 91.43 | 96.00 | 93.66 | 96.26 ± 0.94 |
| | Normal | 97.07 | 97.29 | 97.18 | |
| | Pneumonia | 95.91 | 94.78 | 95.34 | |
| ECOVNet (Hard Ensemble) | COVID-19 | **94.17** | 97.00 | 95.57 | 96.07 ± 0.96 |
| | Normal | 97.05 | 96.72 | 96.89 | |
| | Pneumonia | 94.95 | 94.95 | 94.95 | |
| ECOVNet (Soft Ensemble) | COVID-19 | 92.59 | **100** | **96.15** | 96.07 ± 0.96 |
| | Normal | 97.05 | 96.61 | 96.83 | |
| | Pneumonia | 95.25 | 94.61 | 94.93 | |

Note:
   **Bold** indicates that the method has statistically better performance than other methods for COVID-19.

**Table 7  Class-wise classification results of ECOVNet (Base model EfficientNet-B5) with augmentation.**

| Method | Class | Precision(%) | Recall(%) | F1-score(%) | Accuracy(%)(95% CI) |
|---|---|---|---|---|---|
| ECOVNet (Without Ensemble) | COVID-19 | 87.62 | 92.00 | 89.76 | 94.68± 1.11 |
| | Normal | 97.31 | 94.12 | 95.69 | |
| | Pneumonia | 97.31 | 95.96 | 94.06 | |
| ECOVNet (Hard Ensemble) | COVID-19 | **90.29** | 93.00 | **91.63** | 95.50± 1.02 |
| | Normal | 97.35 | 95.37 | 96.35 | |
| | Pneumonia | 93.76 | 96.13 | 94.93 | |
| ECOVNet (Soft Ensemble) | COVID-19 | 85.45 | **94.00** | 89.52 | 95.19± 1.06 |
| | Normal | 97.67 | 94.92 | 96.28 | |
| | Pneumonia | 93.43 | 95.79 | 94.60 | |

Note:
   **Bold** indicates that the method has statistically better performance than other methods for COVID-19.

ensemble outperforms soft ensemble with a significant margin in the case of precision and F1-score, but an exception is that the soft ensemble is slightly better than the hard ensemble when recall is taken into account. Moreover, in the case of overall accuracy, the hard ensemble shows better detection performance than the soft ensemble. It can also be clearly seen from Tables 6 and 7 that for COVID-19 cases, the precision of the hard ensemble method is better than the soft ensemble method in terms of augmentation and no augmentation. Finally, we also observe that the confidence interval range is small for a no augmentation strategy (Table 6) compared to an augmentation strategy (Table 7).

In Fig. 4, the proposed ECOVNet (base models B0-B5) is aimed at the precision, recall, and F1-score of the soft ensemble of test data considering the COVID-19 cases. When comparing the precision of ECOVNet, we have seen that ECOVNet-B4 (base model B4) shows significantly better performance than the other base models. However, in terms of recall, as we consider more in-depth base models, the value gradually increases. The same is true for F1-scores as well except with a slight decrease of 0.5% from ECOVNet-B5 to ECOVNet-B4.

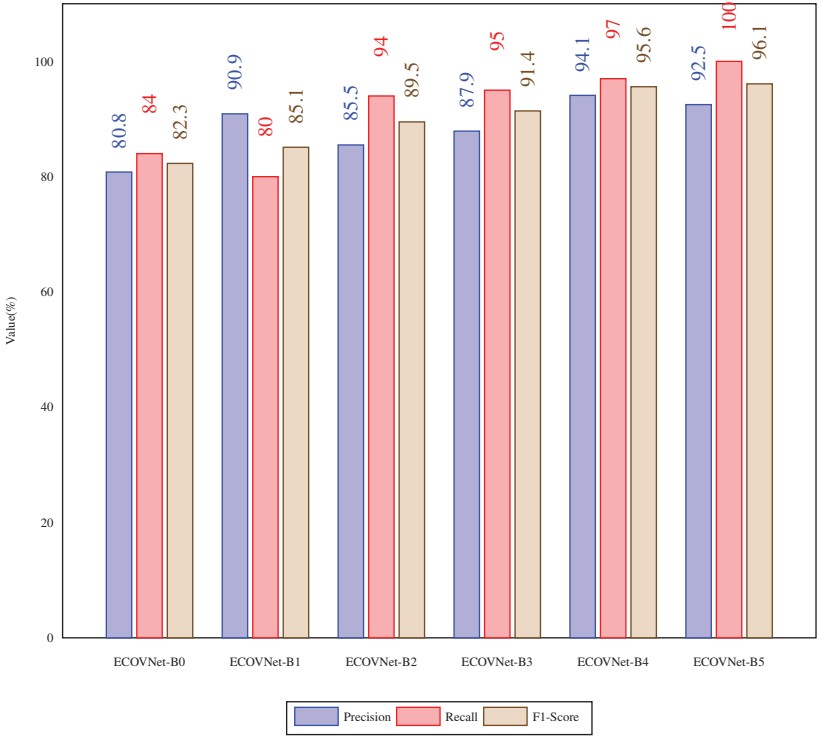

**Figure 4 Precision, Recall, F1 score of the proposed ECOVNet(Base models B0-B5) for test data with soft ensemble considering COVID-19 cases.**

It is often useful to analyze the ROC curve in considering the classification performance of the model since the ROC curve summarizes the trade-off between the true positive rate and the false positive rate of a model that taking into account different probability thresholds. In Fig. 5, the ROC curves show the micro and macro average and class-wise AUC scores obtained by the proposed ECOVNet, where each curve refers to the ROC curve of an individual model snapshot. The AUC scores of all categories are consistent, indicating that the prediction of the proposed model is stable. However, the AUC scores in the third and fourth snapshots are better than other snapshots. As it is evident from Fig. 5 that the area under the curve of all classes is relatively similar, but COVID-19's AUC is higher than other classes, i.e., 1. Furthermore, Fig. 6 shows the confusion matrices of the proposed ECOVNet model considering the base model of EfiicientNet-B5. In Fig. 6, it is clear that for COVID-19, the ensemble methods provide better results than those without ensemble methods. These methods provide results that are 3–4% better than without ensemble. However, ECOVNet can detect normal and pneumonia chest X-rays, whether in ensemble or no ensemble, it can provide the same performance. When comparing normal and pneumonia cases, the proposed ECOVNet model provides excellent results for detecting normal cases. Finally, we can say that ECOVNet is an eminent architecture for detecting COVID-19 cases from chest X-ray images, because it focuses on distinguishing features that help distinguish COVID-19 from normal and pneumonia.

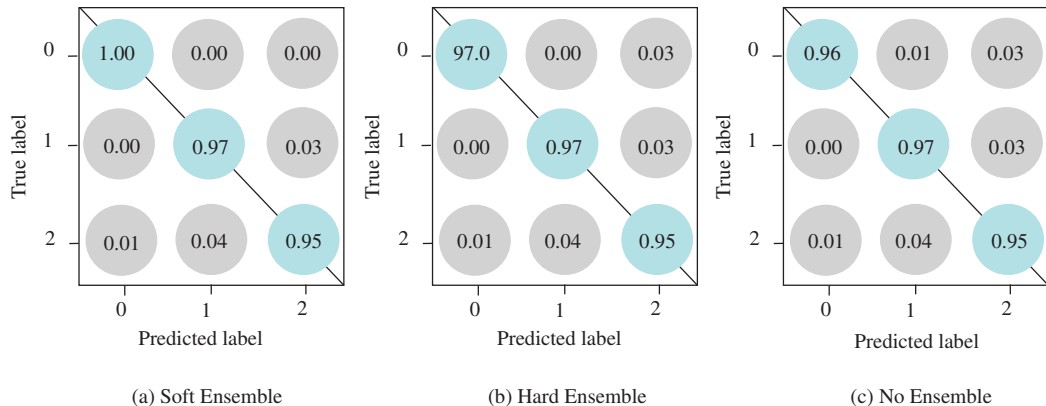

**Figure 5 ROC curves of model snapshots of the proposed ECOVNet considering EfficientNet-B5 as a base model.** (A) to (E) shows ROC curve of each model snapshot.

(a) Soft Ensemble     (b) Hard Ensemble     (c) No Ensemble

**Figure 6 Confusion matrices of the proposed ECOVNet considering EfficientNet-B5 as a base model.** (A) and (B) show confusion matrices for soft and hard ensemble, and (C) is for without an ensemble. In the confusion matrices, the predicted labels, such as COVID-19, Normal, and Pneumonia, are marked as 0, 1 and 2, respectively.

**Table 8 Comparison of the proposed ECOVNet with other state-of-the-art methods on COVID-19 detection.**

| Method | Precision (%) (COVID-19) | Recall (%) (COVID-19) | Accuracy (%) |
|---|---|---|---|
| COVID-Net (*Wang, Lin & Wong, 2020*) | 92.80 | 91.00 | 93.92 |
| EfficientNet-B3 (*Luz et al., 2020*) | 95.29 | 81.00 | 94.49 |
| DarkCovidNet (*Ozturk et al., 2020*) | **96.00** | 88.00 | 92.00 |
| CoroNet (*Khan, Shah & Bhat, 2020*) | 91.00 | 87.00 | 88.00 |
| ECOVNet-Hard Ensemble (Proposed) | 94.17 | 97.00 | **96.07** |
| ECOVNet-Soft Ensemble (Proposed) | 92.59 | **100** | **96.07** |

Note:
**Bold** indicates that the method has statistically better performance than other methods.

## Comparison between ECOVNet and the other models

In comparing ECOVNet with other methods, we considered whether the existing methods used ImageNet weights, applied ensemble methods or used the COVIDx dataset. Table 8 shows the comparison between our proposed ECOVNet method and the state-of-the-art methods for detecting COVID-19 from chest X-rays. We compare our proposed method with COVID-Net (https://github.com/lindawangg/COVID-Net), EfficientNet-B3 (https://github.com/ufopcsilab/EfficientNet-C19), DarkCovidNet (https://github.com/muhammedtalo/COVID-19) and CoroNet (https://github.com/drkhan107/CoroNet). All these methods have released either the trained model (such as COVID-Net), on our used train dataset or the source code. We compare the precision, recall and accuracy of our proposed method with the existing methods using the same training and test dataset (see Table 3) derived from COVIDx note1 While comparing with COVID-Net (*Wang, Lin & Wong, 2020*), we observe that the recall of our proposed method is 6% higher than for COVID-Net. Another method called EfficientNet-B3 (*Luz et al., 2020*) shows higher precision than ours, but its recall and accuracy lag. A method called DarkCovidNet (*Ozturk et al., 2020*) achieves a precision of 96.00%, which is higher than our precision. The proposed ECOVNet is superior to DarkCovidNet in recall and accuracy. The precision, recall and accuracy of CoroNet (*Khan, Shah & Bhat, 2020*) are significantly lower than for our model. As we have observed from the empirical evaluation of the test dataset, the proposed method shows the same classification accuracy in different combinations of soft and hard ensembles. When comparing the results of the soft and hard ensemble, we observed that the soft ensemble showed impressive results when classifying COVID-19 with 100% recall. We have also conducted experiments on EfficientNet, which mainly consists of feature extraction, a global average pool of generated features, and a classification layer. On the other hand, the proposed ECOVNet has considered the use of transfer learning in combination with fine-tuning steps and ensemble methods, so significant improvements can be achieved. In Table 9, it can be seen that our proposed method shows better results than the EfficientNet model.

## Visualization using Grad-CAM

We applied the Grad-CAM visual interpretation method to visually depict the salient areas where ECOVNet emphasizes the classification decision for a given chest X-ray image.

**Table 9 Comparison of the proposed ECOVNet with other CNN architectures on COVID-19 detection.**

| Method | Precision (%) (COVID-19) | Recall (%) (COVID-19) | Accuracy (%) |
|---|---|---|---|
| EfficientNet-B5 (Without augmentation) | **94.12** | 96.00 | 95.76 |
| EfficientNet-B5 (With augmentation) | 84.55 | 93.00 | 95.00 |
| ECOVNet-Hard Ensemble (Proposed) | **94.17** | 97.00 | **96.07** |
| ECOVNet-Soft Ensemble (Proposed) | 92.59 | **100** | **96.07** |

**Note:**
    **Bold** indicates that the method has statistically better performance than other methods.

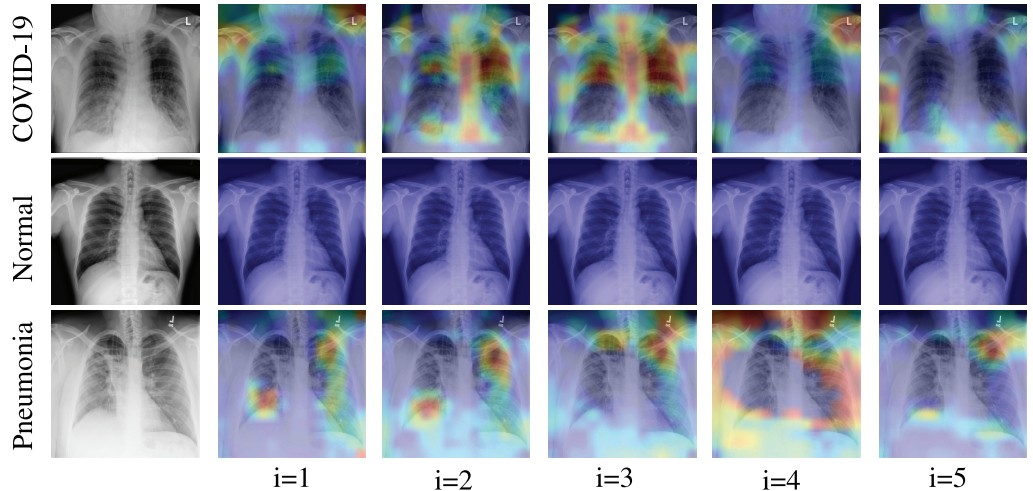

**Figure 7 Grad-CAM visualization for ith snapshot model of the proposed ECOVNet considering the base model EfficientNet-B5.**

Accurate and definitive salient region detection is crucial for the analysis of classification decisions as well as for assuring the trustworthiness of the results. In order to locate the salient area, the feature weights with various illuminations related to feature importance are used to create a two-dimensional heat map and superimpose it on a given input image. Figure 7 shows the visualization results of locating Grad-CAM using ECOVNet for each model snapshots. This salient area locates the area of each category area in the lung that has been identified when a given image is classified as COVID-19 or normal or pneumonia. As shown in Fig. 7, for COVID-19, a ground-glass opacity (GGO) occurs along with some consolidation, thereby partially covering the markings of the lungs. Hence, it leads to lung inflammation in both the upper and lower zones of the lung. When examining the heat maps generated from the COVID-19 chest X-ray, it can be distinguished that the heat maps created from snapshot 2 and snapshot 3 points to the salient area (such as GGO). However, in the case of the normal chest X-ray, no lung inflammation is observed, so there is no significant area, thereby it is easily distinguishable from COVID-19 and pneumonia. As well, it can be observed from the chest X-ray for pneumonia that there are GGOs in the middle and lower parts of the lungs. The heat maps generated for the pneumonia chest X-ray are localized in the salient regions with GGO, but for the 4th snapshot model, it appears to fail to identify the salient regions as the heat

map highlights outside the lung. Accordingly, we believe that the proposed ECOVNet provides sufficient information about the inherent causes of the COVID-19 disease through an intuitive heat map, and this type of heat map can help AI-based systems interpret the classification results.

## CONCLUSION AND FUTURE WORK

In this paper, we proposed a novel modular architecture ECOVNet based on CNN, which can effectively detect COVID-19 with the class activation maps from one of the largest publicly available chest X-ray dataset, i.e., COVIDx. In this work, a highly effective CNN structure (such as the EfficientNet base model with ImageNet pre-trained weights) is used for feature extraction, while fine-tuned pre-trained weights are considered for related COVID-19 detection tasks. Also, ensemble predictions improve the performance by exploiting the predictions obtained from the proposed ECOVNet model snapshots. The results of our empirical evaluations show that the soft ensemble of the proposed ECOVNet model snapshots outperforms the other state-of-the-art methods. Finally, we performed a visualization study to locate significant areas in the chest X-ray through the class activation map for classifying the chest X-ray into its expected category. Thus, we believe that our findings could make a useful contribution to the detection of COVID-19 infection and the widespread acceptance of automated applications in medical practice.

While this work contributes to reduce the effort of health professional's radiological assessment, our future plan is to lead this work to design a fully-functional application using guidelines of the design research paradigm (*Miah & Gammack, 2014*; *Miah, 2008*). Such a modern methodological lens could offer further directions both for developing innovative clinical solutions and associative knowledge in the body of relevant literature.

### Funding
The authors received no funding for this work.

### Competing Interests
The authors declare that they have no competing interests.

### Author Contributions
- Nihad Karim Chowdhury conceived and designed the experiments, performed the experiments, analyzed the data, performed the computation work, prepared figures and/or tables, authored or reviewed drafts of the paper, and approved the final draft.
- Muhammad Ashad Kabir conceived and designed the experiments, analyzed the data, prepared figures and/or tables, authored or reviewed drafts of the paper, and approved the final draft.
- Md. Muhtadir Rahman performed the experiments, analyzed the data, performed the computation work, prepared figures and/or tables, authored or reviewed drafts of the paper, and approved the final draft.

• Noortaz Rezoana performed the experiments, analyzed the data, performed the computation work, prepared figures and/or tables, and approved the final draft.

## Data Availability

The code and raw results are available at GitHub: https://github.com/nihad8610/ECOVNet.

The data is available at GitHub: https://github.com/lindawangg/COVID-Net.

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
