# Peer review of "ECOVNet: a highly effective ensemble based deep learning model for detecting COVID-19"

_PeerJ Computer Science, doi:10.7717/peerj-cs.551_

## Round 0.1 · original submission · Major Revisions

I have received reviews of your manuscript from two scholars who are experts on the cited topic. They find the topic very interesting; however, some concerns must be addressed regarding experimental results and gaps being filled. These issues require a major revision. Please refer to the reviewers’ comments listed at the end of this letter, and you will see that they are advising that you revise your manuscript. If you are prepared to undertake the work required, I would be pleased to reconsider my decision. Please submit a list of changes or a rebuttal against each point that is being raised when you submit your revised manuscript.

Thank you for considering PeerJ Computer Science for the publication of your research. We appreciate your submitting your manuscript to this journal.

Reviewer 1 ·

Basic reporting

It's well written and easy to understand.

Experimental design

The method is clear and straightforward.

Validity of the findings

I have a major concern regarding the results part. The authors created a (small) balanced subset for testing and claimed better performance on the balanced testing set. I am not convinced by the logic behind this. Balanced testing set is unnatural and has little practical value. We all know that machine learning loves balanced dataset. But reporting a good number using an unnatural dataset is a little bit misleading. It also makes it hard to judge the contribution of this work because the numbers reported by other methods do not use this strategy. I would suggest the authors remove everything related to the "balanced dataset" and only report their results on the original dataset.

Additional comments

1. Some related works are not mentioned, for example: https://doi.org/10.1038/s41467-020-18685-1
2. It's a little misleading to use an unnatural balanced testing dataset. It also makes it hard to compare with other methods unless they are using the same strategy. So I would suggest the authors to remove everything related to the balanced dataset and report results on the original dataset.

Reviewer 2 ·

Basic reporting

The paper is well written and organized.

The literature references about work related to Covid are sufficient, but there are no references for the vast amount of deep learning work in other closely related areas. There are also many references about well known concepts like ensemble methods.

The article is well structured, and the data is publicly available.

The paper contains excessively lengthly explanations of well known concepts like ensemble learning and class activation maps.

The results sections is satisfactory.

Experimental design

The research cannot be said to be original. There is no identified knowledge gap being filled. Applying ensemble methods is not sufficient to meet this requirement.

The method applies a stock EfficientNet with the final layers changed from 1000 classes (ImageNet) to 3 classes. Two fully connected layers are added, but there is no comparison included to show that these contribute to performance.

The size of the validation and test sets is too small with 90% of the data used for training. There are only a very small number of positive Covid cases in the validation and test sets.

Methods are well described and replicable.

Validity of the findings

The data is provided and the results could be replicated.

The work does not answer any research question, as it applies stock methods with little to no modification to a very small dataset. The paper claims a very minor modification to a stock EfficientNet as a novel CNN architecture. As such, it is not clear what conclusions can be made from the presented research.

---

## Round 0.2 · Minor Revisions

This resubmission still requires revisions. It is of paramount importance to check the experimental results section, specifically Table 8. If you are prepared to undertake the work required, I would be pleased to reconsider my decision. Please submit a list of changes or a rebuttal against each point that is being raised when you submit your revised manuscript.

Reviewer 1 ·

Basic reporting

no comment

Experimental design

no comment

Validity of the findings

no comment

Additional comments

I would say that the revised version is much improved. I still have two major concerns regarding table 8 which should be addressed before publishing.

1. Table 8 compared the proposed method with other existing methods so its accuracy is very important. I haven't checked all numbers but at least for DeepCOVIDExplainer the number in Table 8 does not match the one I found in either https://ieeexplore.ieee.org/document/9313304 or https://arxiv.org/pdf/2004.04582.pdf. Perhaps the authors of DeepCOVIDExplainer have updated their numbers, but it is still better for the current work to cite numbers from the peer-reviewed version (the IEEE version) or at least the latest pre-print version instead of the old pre-print version. Please check the numbers of other methods too to make sure all numbers are current and accurate.

2. It is not explained in the text why COVID-Net, EfficientNet-B3, DeepCOVIDExplainer, and the proposed method all use the COVIDx dataset but have different dataset size in Table 8. It would be much better if the authors could compare the proposed method with other methods on the same dataset, at least for methods that use COVIDx, if possible.

---

## Round 0.3 · accepted · Accept

I am pleased to inform you that your work has now been accepted for publication in PeerJ Computer Science.

Thank you for submitting your work to this journal.

Reviewer 1 ·

Basic reporting

no comment

Experimental design

no comment

Validity of the findings

no comment

Additional comments

All my concerns have been successfully addressed by the authors.